# The Kinematics and Dynamics of *Schizopygopsis malacanthus* Swimming during *U_crit_* Testing

**DOI:** 10.3390/ani12202844

**Published:** 2022-10-19

**Authors:** Yangxi Li, Yiqun Hou, Ben Zhang, Xuan Zou, David Johnson, Fan Wan, Chaoyan Zhou, Yao Jin, Xiaotao Shi

**Affiliations:** 1Key Laboratory of Ecological Impacts of Hydraulic-Projects and Restoration of Aquatic Ecosystem of Ministry of Water Resources, Institute of Hydroecology, Ministry of Water Resources and Chinese Academy of Sciences, Wuhan 430079, China; 2College of Hydraulic and Environmental Engineering, China Three Gorges University, Yichang 443002, China; 3Shanghai Survey and Design Research Institute Co., Ltd., Shanghai 200000, China; 4GNSS Research Center, Wuhan University, 129 Luoyu Road, Wuhan 430079, China; 5School of Natural Sciences and Mathematics, Ferrum College, Ferrum, VA 24088, USA; 6Northwest Engineering Corporation Limited, Xi’an 710065, China

**Keywords:** swimming kinematics, swimming dynamics, locomotive behavior, *U_crit_*, energy consumption

## Abstract

**Simple Summary:**

The swimming ability of fish plays a guiding role in the construction of fish passing facilities. However, it is not enough to know the swimming ability of fish. We need to know the behavior and energy consumption of fish in the process of movement, so as to determine the activity area and migration suitable for fish. Therefore, a video tracking program was used to record and analyze the motion of five test fish in a Brett-type flume during each velocity step. The results obtained the kinematic and dynamic parameters of fish. Secondly, we found that steady fish swimming is not entirely steady in flowing water, with swimming speed varying by 2.2% to 8.4% and increasing with flow velocity. Further, because energy expenditure increases with the cube of swimming speed, a slight excess in fish passage flow velocity could result in a disproportionately large decrease in the rate of passage success. Therefore, we recommend guarding against an excessive flow velocity in the main velocity zone and ensuring that resting pools along the passageway are adequate. Our characterization of the kinematics and dynamics of fish swimming provides important new information to consider when indices of swimming ability from controlled tank testing are applied to fish passage design.

**Abstract:**

The swimming kinematics (how fish move) and dynamics (how forces effect movement) of *Schizopygopsis malacanthus* were investigated during the determination of $U_{crit}$ by stepped velocity testing. A video tracking program was used to record and analyze the motion of five test fish in a Brett-type flume during each velocity step. The findings fell into three groups: (1) Even when flow was uniform, fish did not swim steadily, with speeds fluctuating by 2.2% to 8.4% during steady swimming. The proportion of unsteady swimming time increased with water velocity, and defining steady and unsteady swimming statistically, in terms of the definition of standard deviation of instantaneous displacements, may have higher accuracy. (2) In steady swimming, the forward velocity and acceleration of fish were correlated with body length (*p* < 0.05), but in unsteady swimming the correlations were not significant. The maximum swimming speed (1.504 m/s) and acceleration (16.54 m/s^2^) occurred during unsteady swimming, but these measurements may not be definitive because of tank space constraints on fish movement and the passive behavior of the test fish with respect to acceleration. (3) Burst-coast swimming in still water, investigated by previous scholars as an energy conserving behavior, is not the same as the gait transition from steady to unsteady swimming in flowing water. In this study, the axial force of fish swimming in the unsteady mode was significantly higher (×1.2~1.6) than in the steady mode, as was the energy consumed (×1.27~3.33). Thus, gait transition increases, rather than decreases, energy consumption. Our characterization of the kinematics and dynamics of fish swimming provides important new information to consider when indices of swimming ability from controlled tank testing are applied to fish passage design.

## 1. Introduction

Flumes with continuously circulating water are commonly used for testing fish swimming performance, particularly to measure the steady swimming speed of fish [1]. A stepped velocity test, in which the flow velocity in the flume increases by prescribed steps at prescribed time intervals, is currently the most frequently used method. The critical swimming speed ($U_{crit}$) is used to examine the impact of environmental conditions on fish [2], to calculate fish physiological energy consumption [3], and to establish design criteria for fish passage flow rates [4]. However, very little research has been carried out on the kinematics and dynamics of fish swimming during $U_{crit}$ testing.

Fish swimming is generally classified as steady swimming and unsteady swimming. The critical swimming speed ($U_{crit}$) is commonly employed as an evaluation index for steady swimming in which the swimming speed and direction are nearly constant [5]. Unsteady swimming, which includes quick starts, rapid acceleration [6], turns [7], and burst-coast [8], is a more complicated style of swimming, typically of brief duration during rapid changes in speed and direction. Gait transitions, or the change from steady swimming to burst-coast swimming, occur often during fish locomotion in nature [9,10], and gait change has been previously examined. For example, Weihs [11] postulated that, from an energetics standpoint, fish can conserve 46% of their energy using burst-coast motion. Wu et al. [12] used particle image velocimetry (PIV) to reveal the kinematic characteristics and wake patterns of carp burst-coast swimming in an open tank with still water, concluding that burst-coast motion saves 45% of energy compared to steady swimming. However, Yang et al. [13] found that when the water velocity in an open tank exceeds 0.34 m/s, burst-coast swimming does not reduce energy consumption. We observed similar gait changes during the $U_{crit}$ test, and with the development of new animal behavior observation techniques, a more detailed and quantitative study of fish swimming behavior is now possible. However, the fine motions of fish in a $U_{crit}$ flume test have yet to be examined. Our objective is to clarify the kinematics and dynamics of fish swimming in a closed flume.

A riverine fish, *Schizopygopsis malacanthus*, was chosen as the test fish to examine the fine motions of fish swimming over a range of flow velocities in a circular tank test, because it lives in the flowing water and its size is suitable for testing in the loligo tank. Kinematic and kinetic parameters, including displacement, velocity, acceleration, axial force, and energy consumption, were quantized, and movement at 1/30-s intervals was analyzed during steady and unsteady swimming. Behavioral characteristics and swimming patterns of test fish swimming at the different flow velocities were analyzed to explore the influence of the flume on fish swimming behavior. Our findings provide additional information to help guide the interpretation of swimming velocity indices obtained in Brett-type swimming flumes.

## 2. Materials and Methods

### 2.1. Experimental Materials

*S. malacanthus* (*n* = 5, weight = 34.8~152.9 g, body length = 14.3~18.3 cm) were obtained by seining in the Dadu River, Xinzagou, Jinchuan County, Ganzi Tibetan Autonomous Prefecture, Sichuan Province, China in early May 2020 (Figure 1). The captured fish were maintained in a pool with a water exchange flow rate of 0.05 L/s and dissolved oxygen maintained at ≥6.0 mg/L. The water temperature in the test flume was approximately the same as in the holding pool, ranging from 11.9 to 14 °C. Test fish were not fed for 24 h prior to testing.

### 2.2. Critical Swimming Ability Evaluation

As shown in Table 1, a stepped velocity test was carried out to determine the critical swimming speed ($U_{crit}$) of *S. malacanthus* (*n* = 5), and each fish was tested only once. The test flume was an elliptical tank (L × W × H = 70 cm × 20 cm × 20 cm) from the Danish company (Figure 2), LoligoSystem (Model SW10200). The initial flow velocity was 0.4 m/s and was increased by 0.2 m/s at 15 min intervals until the fish was fatigued (i.e., the fish could not continue swimming against the flow, rested against the downstream net, and did not resume swimming after tapping the downstream wall of the tank for 20 s). Time to fatigue, critical swimming speed, and body weight and length were recorded. The entire process was recorded with a Sony camcorder at 30 fps. $U_{crit}$ was calculated using Equation (1) [14]:(1)$$U_{crit}=U_{t}+\frac{t}{\Delta t}\Delta U$$
where $U_{t}$ is the maximum swimming speed (cm/s) the test fish maintained for the entire time interval; Δt is the time interval (15 min); ΔU is the flow rate increment (0.2 m/s), and t is the time elapsed in the uncompleted interval at fatigue.

### 2.3. Calculation of Kinematic Parameters for S. malacanthus

In the next four subsections Section 2.3, Section 2.4, Section 2.5, Section 2.6 numerous symbols are used, and they are listed and defined in Table 2.

Due to the software’s processing capability and the huge amount of data needed for microscopic analysis, video clips of the middle 20 s at each flow velocity increment were analyzed frame by frame using the motion tracking software Etho Vision XT 9.0 [15]. The results of other time periods are comparable to this results, and as per the study dynamics of Wu et al. [12], Yang et al. [13], and Ashraf et al. [16], only the data in a period of 0.4 s, 0.4 s, 10 s were analyzed. The center-of-mass co-ordinates of fish were obtained at 1/30-s intervals, and the instantaneous displacement, instantaneous ground velocity, and instantaneous acceleration at 1/30-s intervals were calculated from the change in co-ordinates. Because the motion of fish against water flow was the main consideration, vertical motion was not analyzed. The calculation procedures are described below.

Location:

X-coordinate of frame 1: $x_{0}$

X-coordinate of frame 2: $x_{1}$

X-coordinate of frame *i* + 1: $x_{i}$

Displacement:

The instantaneous displacement of fish at 1/30 s intervals over 20 s is:$$S_{0}=(x_{1}-x_{0}),\;\;S_{1}=(x_{2}-x_{1})S_{2}=(x_{3}-x_{2}),\;\ldots ,\;S_{i}=(x_{i}-x_{i-1})$$

Velocity:

A positive instantaneous velocity value is assigned to fish moving upstream, i.e., a ‘forward velocity’, recorded as $v_{c}$, and a backward instantaneous velocity value is assigned to a fish moving downstream, i.e., a ‘backward velocity’, recorded as $v_{s}$. The instantaneous ground velocity of fish at 1/30 s intervals over a 20-s period is:$$v_{1}=\frac{\left(s_{1}-s_{0}\right)}{1/30},\;v_{2}=\frac{\left(s_{2}-s_{1}\right)}{1/30},\;\ldots ,\;v_{i}=\frac{s_{i}-s_{i-1}}{1/30}$$

$U_{max}$ is defined as the maximum swimming speed (*v*) of each fish at a given flow velocity increment.

Acceleration:

A positive instantaneous acceleration is assigned to fish moving upstream, i.e., ‘forward acceleration’, recorded as $a_{c}$, and a backward instantaneous acceleration is assigned to a fish moving downstream, i.e., a ‘backward acceleration’, recorded as $a_{s}$. The instantaneous acceleration of fish at 1/30 s intervals over 20 s is: $$a_{1}=\frac{\left(v_{2}-v_{1}\right)}{1/30},\;a_{2}=\frac{\left(v_{3}-v_{2}\right)}{1/30}\ldots ,\;a_{i-1}=\frac{v_{i}-v_{i-1}}{1/30}$$

$a_{max}$ is defined as the maximum acceleration of each fish at a given flow velocity increment.

### 2.4. Classification of Steady and Unsteady Swimming Modes

The standard deviation of the instantaneous displacements of each fish within the 20-s video clip was determined and a ‘steady period’ was defined as a time interval when the instantaneous displacement was within one standard deviation of the mean displacement, denoted as $t_{w}$, and the corresponding motion was classified as steady swimming. An ‘unsteady period’, denoted as $t_{f}$, was defined as a time interval when the instantaneous displacement exceeded a single standard deviation from the mean, and the corresponding motion was classified as unsteady swimming. Thus, steady and unsteady swimming are classified as follows:

Average instantaneous displacement: $\bar{S}$=$\frac{s_{0}+s_{1}+\ldots +s_{i}}{i+1}$

Standard deviation of instantaneous displacement: $\sigma =\sqrt{\frac{\sum_{i=1}^{n}(S_{i}-\bar{S}{)}^{2}}{n}}$

Steady mode: $-\sqrt{\frac{\sum_{i=1}^{n}S_{i}-{\bar{S}}^{2}}{n}}<S_{i}-\bar{S}<\sqrt{\frac{\sum_{i=1}^{n}(S_{i}-\bar{S}{)}^{2}}{n}}$

The displacement due to steady swimming is the average of all instantaneous displacements in this range and displacement by unsteady swimming is calculated in a similar manner using the equations below:


$$\mathrm{Unsteady}\;\mathrm{mode}:S_{i}-\bar{S}<-\sqrt{\frac{\sum_{i=1}^{n}S_{i}-{\bar{S}}^{2}}{n}},\mathrm{or}\;S_{i}-\bar{S}>-\sqrt{\frac{\sum_{i=1}^{n}S_{i}-{\bar{S}}^{2}}{n}}$$


### 2.5. Calculation of Swimming Kinetic Parameters

The increase in axial force must be sufficient to accelerate the fish mass and to push the surrounding fluid out of the path, referred to as ‘added mass’ [17]. The axial force generated consists of two components: the force to overcome the drag on the fish body (*f*), and the force to accelerate the mass of the body and the fluid around it (*F*). The total axial force ($F_{total}$) generated by fish in the steady and unsteady modes is calculated according to Equation (2).
(2)$$F_{total}=\int_{0}^{t_{i}}F_{i}+f_{i}dt$$

Equations (3)–(8), used to calculate *F* and *f*, are detailed below:(3)$$F_{i}=C_{A}ma_{i}$$
where *F_i_* is the force to accelerate the mass of the body and the fluid around it in 1/30-s time increments, $C_{A}$ is the dimensionless added mass coefficient (related to object shape, ~1.0 for streamlined bodies, [18], m is fish mass (g), and $a_{i}$ is the dynamic acceleration (m/s^2^).
(4)$$f_{i}=0.5C_{d}\rho A_{s}U_{i}{}^{2}$$
where *f_i_* is the drag on the fish in 1/30-s time increments. $C_{d}$ is the drag coefficient and consists of the friction coefficient $C_{f}$ and the pressure coefficient $C_{p}$ [19]. $C_{d}$ is generally considered to be ~1.2 $C_{f}$ 3 and, thus, can be calculated using Equation (5).
(5)$$C_{d}=C_{f}+C_{p}\approx 1.2C_{f}$$

$C_{f}$ is related to the flow state (laminar or turbulent) and can be calculated from the Reynolds number, *Re*. Flow in the flume is laminar and $C_{f}$ can be calculated using Equation (6) [20].
(6)$$C_{f}=0.072Re^{-0.2}$$

The Reynolds number is calculated using Equation (7):(7)$$Re=\frac{L\cdot U}{\vartheta }$$
where *L* the total fish length (cm), *U* is the average swimming speed (m/s), and ϑ (m^2^/s) is the kinematic viscosity of water, calculated using Equation (8) [21]:(8)$$\vartheta =\frac{1.775\times 10^{-6}}{1+0.0337T+0.000221t^{2}}$$
where T is water temperature in the flume (°C) and $t_{i}$ refers to the time spent in steady or unsteady swimming, as calculated in the previous section.

Finally, to complete the definition of terms in Equation (4): ρ is the density of water (1000 kg/m^3^), *A_s_* is the wetted surface area of the fish ($A_{S}=\alpha L_{f}{}^{\beta },\;\mathrm{where}\;L_{f}$ is fish body length, and  α=0.465  and β=2.11  are empirical coefficients) and $U_{i}$ is the instantaneous absolute swimming speed ($U_{i}=v_{i}+v_{w}$, where $v_{w}$ is water velocity and $v_{i}$ is the instantaneous fish velocity as previously defined).

### 2.6. Calculation of the Energy Consumed by Swimming

There are currently two methods to understand the energy consumption of fish through the fishway. One is to estimate the drag forces of the fish through the flow rate [22], the second is to measure the oxygen consumption rate of the fish in a closed tank [23]. We choose the second method, and the energy consumed is equivalent to the work done by the motion of swimming fish, i.e., work = force × distance [24]. Swimming distance was calculated separately for the time periods of steady and unsteady swimming during the 20-s intervals for each flow velocity. The estimated swimming distance of fish during steady and unsteady mode swimming was calculated as duration (t) × water velocity (*v_w_*). Steady swimming and unsteady swimming energy consumption are both the product of the total axial force (*F_total_* = *f* + *F*) and displacement (*S*), as expressed in Equation (9).
(9)$$E_{total}=\frac{S_{1}(f_{1}+F_{1})+S_{2}(f_{2}+F_{2})+\ldots +S_{i}(f_{i}+F_{i})}{v_{w}t_{i}}$$

### 2.7. Data Analysis

The data were further processed using Excel 2019 and all graphs were produced using Origin 2019. One-way analysis of variance (ANOVA) was used to test correlations between time share, velocity, acceleration, axial force, energy consumption, and flow velocity in the steady and unsteady modes. A linear correlation function was fitted to the data using SPSS 25.0. Statistical values are expressed using the mean ± standard error (*M* ± *SE*) and statistical significance was set at *p* < 0.05.

## 3. Results and Analysis

### 3.1. Critical Swimming Speed

The $U_{crit}$ for each test fish and the mean value (0.86 ± 0.06 m/s) are shown in Table 3. The video recorded at a flow velocity of 1.0 m/s was not included in the kinetic analysis as only one fish completed the 15-min interval. Two test fish started, but did not complete, the 1.0 m/s increment (0.8 m/s < $U_{crit}$ < 1.0 m/s) and two fish were fatigued during the 0.8 m/s increment ($U_{crit}$ ≤ 0.8 m/s).

### 3.2. Swimming Mode Times (t_w_, t_f_) and Their Relationship to Water Velocity (v_w_)

The swimming mode was determined by analyzing video recordings of *S. malacanthus* swimming at water velocities of 0.4 m/s, 0.6 m/s, and 0.8 m/s, as described in Section 2.4. The ranges, mean values, standard deviations of instantaneous displacements (*S*), and the physical indicators used to analyze *S. malacanthus* swimming at each water velocity are summarized in Table 4.

As shown in Figure 3, the proportion of time that test fish swam in the steady mode ranged from 70.04% (*v_w_* = 0.8 m/s) to 85.07% (*v_w_* = 0.4 m/s). The percentage of steady mode swimming time of *S. malacanthus* negatively correlated with flow velocity and the relationship was linear: *y* = −0.3714*x* + 0.9975 (*R*^2^ = 0.9996, *p* < 0.05). Conversely, the fraction of time swimming in the unsteady mode increased with water velocity, from 14.93% to 29.96%, and the linear correlation was positive: *y* = 0.3714*x* + 0.00255 (*R*^2^ = 0.9996, *p* < 0.05).

### 3.3. Correlation between Swimming Speed, Acceleration, and Water Velocity in the Steady and Unsteady Modes

As shown in Figure 4, video analysis indicates that the *S. malacanthus* body still slightly fluctuates during steady swimming. With each tailbeat, there is a slight acceleration and deceleration of the fish. The forward velocity ($v_{c}$) of fish under steady swimming ranged from 0.022~0.084 (0.055 ± 0.012) m/s. The fluctuation of fish swimming speed ranged from 2.2% to 8.4% and increased with water flow velocity, as shown in Figure 3. The backward velocity ($v_{s}$) decreased with increasing water velocity and ranged from −0.083~−0.021 (−0.054 ± 0.02) m/s. The $v_{c}$ and $v_{s}$ of *S. malacanthus* ranged from 0.078~0.194 (0.144 ± 0.05) m/s and −0.186~−0.105 (−0.146 ± 0.05) m/s, respectively.

As shown in Figure 5, the forward acceleration $a_{c}$ of *S. malacanthus* in steady mode ranged from 0.9~3.012 (2.224 ± 1.036) m/s^2^, which was positively correlated with the water velocity. The backward acceleration $a_{s}$ decreased with increasing water velocity and ranged from −3.203~−0.797 (−2.258 ± 1.106) m/s^2^. The $a_{c}$ and $a_{s}$ of *S. malacanthus* in the unsteady mode ranged from 2.012~5.079 (3.557 ± 0.89) m/s^2^ and −4.92~−2.258 (−3.602 ± 0.829) m/s^2^, respectively.

### 3.4. Comparison of Axial Force and Energy Consumption in Steady/Unsteady Mode

We calculated the axial forces generated by *S. malacanthus* in two movement modes for 20 s at each flow gradient based on empirical equations. The axial forces in the steady mode ranged from 0.078~0.36 (0.19 ± 0.066) N. The axial forces in the unsteady mode were significantly higher than those in the steady mode and were 1.17~1.64-times higher than those in the steady mode (*p* < 0.05), ranging from 0.128~0.397 (0.249 ± 0.079) N.

As shown in Figure 6, the energy consumed by fish in the unsteady mode ranged from 0.030~0.047 (0.039 ± 0.016) J/m, which was significantly higher than that consumed in the steady mode and was 1.27~3.33-times higher than that consumed in the steady mode (*p* < 0.05). The range of energy consumption in the steady mode was 0.009~0.037 (0.023 ± 0.016) J/m.

## 4. Discussion

### 4.1. Steady and Unsteady Swimming Modes during the $U_{crit}$ Test

It is generally accepted that the $U_{crit}$ determined in the Brett flume is the maximum steady swimming speed of fish [19]. In this study, we found that unsteady swimming occurs at flow velocities below $U_{crit}$. The unsteady mode at low flow velocities may be related to the fish tailbeat and to the criteria used in this study to distinguish between steady and unsteady swimming. In fact, the swimming speed fluctuations produced by fish tailbeating has been reported by previous scholars. Observations of swimming eels have noted fluctuations about the mean velocity ranging from 4% [25] to 10% [26], and swimming mullets have been found to exhibit velocity fluctuations exceeding 20% [27,28]. In this study, the fluctuations in steady swimming velocity of *S. malacanthus* varied from 2.2% to 8.4% and increased with water velocity. While fish shape, kinematics, and Reynolds number may all play a role in velocity fluctuations, recent simulation studies have shown that body shape has a greater effect on velocity fluctuations than fish kinematics, i.e., changes in body shape produce larger fluctuations than changes in body kinematics [29].

The steady mode was defined by Wise et al. [30] as a variation in fish motion not exceeding 2% of body length, while incremental displacement (S) not exceeding ±1 SD was employed as the criterion in this study. The displacement of fish in the steady mode at each flow velocity was divided by body length to compare differences between the two criteria (Figure 7), and the displacement of fish movement in the steady mode was obtained between 0.55% and 2.19% of body length. Indeed, our findings show that fish motion does not significantly change in the steady mode at a modest water flow velocity (0.4 m/s) With such a broad definition, even a 2% inaccuracy would not be consequential. Furthermore, the influence of fish size is not considered by the statistical criterion. Therefore, we consider the standard deviation to be a better criterion for defining steady and unsteady swimming modes.

To summarize, the kinematics of swimming produces small fluctuations even in the steady swimming mode, and fish swimming can never be absolutely steady.

### 4.2. Limiting Factors of Fish Acceleration Capacity

The ability of fish to accelerate is crucial for successful escape and predation. Varying swimming speed and acceleration is also important behavioral aspects of fish reproduction and courtship [31]. In the steady mode, fish accelerate and decelerate over small distances to maintain swimming speed within a certain range, and acceleration was positively correlated with water velocity. In addition to the influence of water velocity, fish species and body length also affect acceleration. For example, the acceleration of *Cyprinus carpio* koi (body length 5.25~5.85 cm) was 0.236 ± 0.164 m/s^2^ [12] and that of *Mylopharyngodon piceus* (body length 17.93 ± 1.27 cm) was 7.313 ± 4.233 m/s^2^ in the unsteady mode [32], while the acceleration of *S. malacanthus* (body length 16.42~1.819 cm) in this study was 4.233 ± 1.535 m/s^2^ in the unsteady mode. Fish with a spindle type body form may have stronger acceleration ability since this body shape reduces water resistance, while fish with a laterally flattened shape or rod shape have relatively weak acceleration and less nimble movement.

The maximum swimming speed of fish and their ability to accelerate are also of great ecological importance. Scholars have previously tested the maximum swimming speed of fish using acoustic, electrical, and mechanical stimulation. For example, Lu Bo et al. [33] measured the maximum swimming speed of *Ctenopharyngodon idella* (17.93 ± 1.27 cm) as 2.36 m/s using sound to startle the fish, this was nearly three-times the $U_{crit}$ obtained by Xian et al. [34] using a Brett-type flume. *Mylopharyngodon piceus* and *S. malacanthus* are both carp of similar body size, but the highest swimming speed of *S. malacanthus* pushed to swim at sprint speed in this study by increasing water velocity was 1.50 m/s, only 1.75-times $U_{crit}$. However, in the former case, fish were highly stimulated to accelerate by loud noise or electric shock, triggering escape behavior as when a predator is encountered. In this study, the stimulation of increased water velocity did not trigger a flight response and could explain the difference in results. Further, the flume constricts fish mobility and test fish are forced to swim at a constant speed that may be more tiring; these conditions are potentially inferior to open, still water for testing sprint speed. The accuracy of using flumes to evaluate fish swimming ability has been questioned because the swim channel of circular flumes is so limited and potentially constricts fish movement in the unsteady swimming mode [35]. Our results indicate that, when a swimming flume is used for testing, the data acquired for fish swimming steadily are more reliable than for fish swimming in the unsteady mode.

### 4.3. Thrust and Energy Consumption in the Steady and Unsteady Swimming Modes

Many researchers have studied the axial force generated by fish and the resulting energy consumption. The energy consumption of fish swimming in the flume in the unsteady mode was 1.27~3.33-times greater than in the steady mode, while the axial force was 1.17~1.64-times greater. Denny [36] reasoned that, because the drag force on fish is proportional to the square of the velocity, while energy consumption is proportional to the velocity cubed, fish become fatigued more quickly when swimming in the unsteady mode [37]. On the other hand, Wu et al. [12] demonstrated that unsteady burst-coast swimming uses 45% less energy than swimming steadily. In this study, work was measured during steady and unsteady swimming, and the results indicate that unsteady swimming increases the drag force and energy expenditure compared with steady swimming.

The main reason for this discrepancy is that Wu’s experiment was carried out in still water, and the energy savings of the unsteady mode were mainly from the coast phase of burst-coast swimming. The $C_{d}$ of continuous swimming is the same as for burst swimming (0.242 ± 0.024), much larger than $C_{d}$ in the coast phase (0.060 ± 0.003) that results in an energy savings of 45%. According to Wu’s premise, tailbeating was near continuous throughout the test during both acceleration and deceleration. The $C_{d}$ values (Equation (5)) for steady and unsteady modes at each flow gradient in this investigation were the same because the Reynolds number was calculated (Equation (7)) using the same flow velocity and fish length for both modes and there is no energy-saving coast phase. We conclude that the axial force and energy consumption are greater in the unsteady swimming mode than in the steady mode, consistent with the analysis of Yang et al. [13] and Wise et al. [30].

According to the observational study of Ashraf et al. [16], fish display very little continuous swimming in still water, preferring the burst-coast mode to conserve energy. In the natural world, unlike the conditions in this controlled study, the constant flow velocity necessary for swimming at a uniform speed is not the norm. However, fish species such as *S. malacanthus* inhabit stream segments with water that is fast flowing and seldom still. Under these conditions, fish are observed steadily swimming upstream, rather than the burst-coast swimming mode observed in still water. Therefore, for fish inhabiting fast flowing streams, the $U_{crit}$ determined in a Brett-type flume will reliably indicate the maximum ability of fish to swim against a current. The experimental apparatus used to test fish swimming ability must simulate water flow conditions in the natural habitat of the species being tested.

Migrating fish swim upstream for long distances, and current thinking on fish passage design is to have an upstream flow channel with a clear direction with resting pools as appropriate. Based on the findings of this study, the $U_{crit}$ results obtained using the Brett-type flume provide useful design criteria for flow velocities. Further, excessive flow velocities in fish passages must be avoided as drag forces increase in proportion to the square of swimming speed and energy consumption increases in proportion to the cube of swimming speed. Doubling swimming speed increases energy consumption by a factor of eight. When red muscle energy from aerobic respiration is insufficient to support fish movement, white muscle energy from anaerobic respiration is required and this results in rapid fatigue. Thus, even a slightly excessive velocity can have a large impact on the rate of successful fish passage. Further, if target species inhabit swiftly flowing streams, the mainstream zone of the fishway should have a nearly constant flow velocity so that fish can swim steadily. From a kinetics standpoint, a vertical slit fishway with a distinctly sigmoidal mainstream is also beneficial to fish passage [38].

## 5. Conclusions

We recorded and analyzed fish movement in a circulating flume during the determination of $U_{crit}$, with constant flow velocity at each flow increment. Kinematic and dynamic parameters were measured, and characteristic functions were calculated at each flow velocity; fish displacement, velocity, acceleration, axial force, and energy consumption. We found that fish acceleration may be constrained by the relatively small size of the test flume swim chamber, but that *U_crit_* in the steady swimming mode can be reliably determined. With respect to fish passage design, we found that steady fish swimming is not entirely steady in flowing water, with swimming speed varying by 2.2% to 8.4% and increasing with flow velocity. More importantly, in contrast to earlier investigations on fish burst-coast swimming in still water, we found that gait transition from steady to unsteady swimming in flowing water results in higher, rather than lower, energy expenditure. Further, because energy expenditure increases with the cube of swimming speed, a slight excess in fish passage flow velocity could result in a disproportionately large decrease in the rate of passage success. Therefore, we recommend guarding against an excessive flow velocity in the main velocity zone and ensuring that resting pools along the passageway are adequate.

## Figures and Tables

**Figure 1 animals-12-02844-f001:**
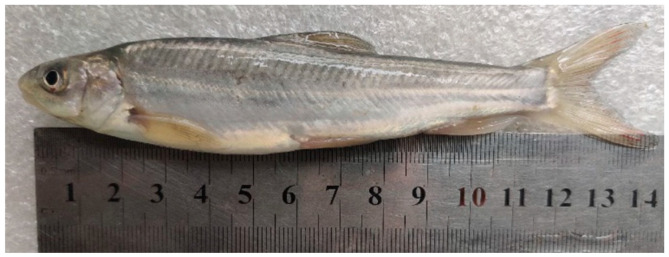
Experimental fish: *Schizopygopsis malacanthus*.

**Figure 2 animals-12-02844-f002:**
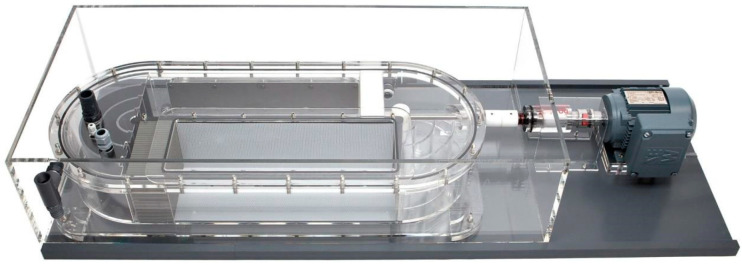
Loligo swim tunnel (https://www.loligosystems.com, (accessed on 10 May 2020)).

**Figure 3 animals-12-02844-f003:**
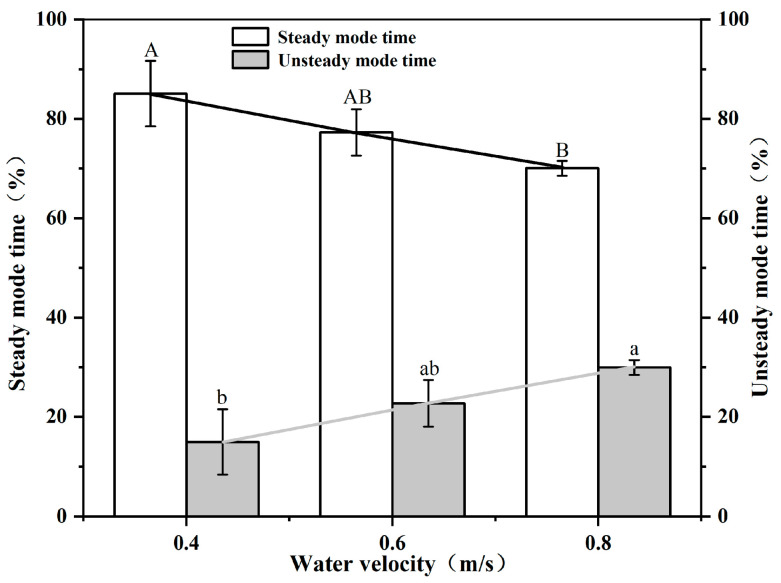
Correlation between steady/unsteady mode time and water flow velocity for *S. malacanthus.* Different upper cases indicate the significant the difference of water velocity in steady mode time treatments; Different lower cases indicate the significant the difference of water velocity in unsteady mode time treatments.

**Figure 4 animals-12-02844-f004:**
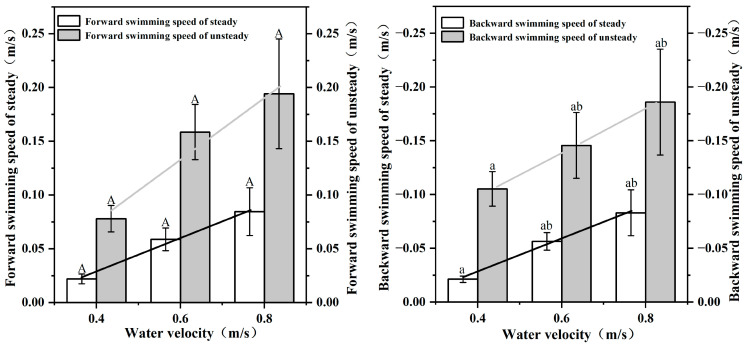
Relationship between forward/backward velocity and water flow velocity in steady/unsteady mode of *S. malacanthus* swimming. Different upper cases indicate the significant the difference of water velocity in forward swimming speed of steady/unsteady treatments; Different lower cases indicate the significant the difference of water velocity in backward swimming speed of steady/unsteady treatments.

**Figure 5 animals-12-02844-f005:**
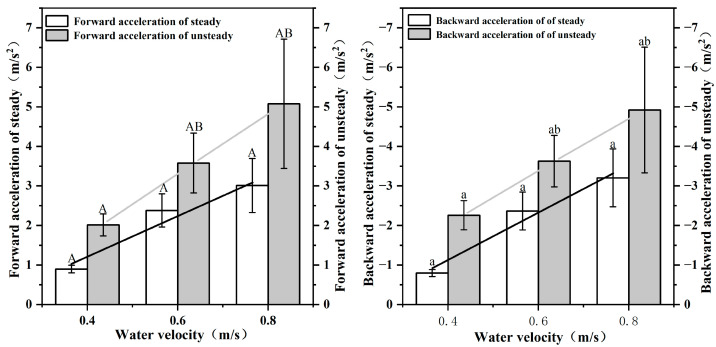
Relationship between forward/backward acceleration and water flow velocity in steady/unsteady mode of *S. malacanthus* swimming. Different upper cases indicate the significant the difference of water velocity in forward acceleration of steady/unsteady treatments; Different lower cases indicate the significant the difference of water velocity in backward acceleration of steady/unsteady treatments.

**Figure 6 animals-12-02844-f006:**
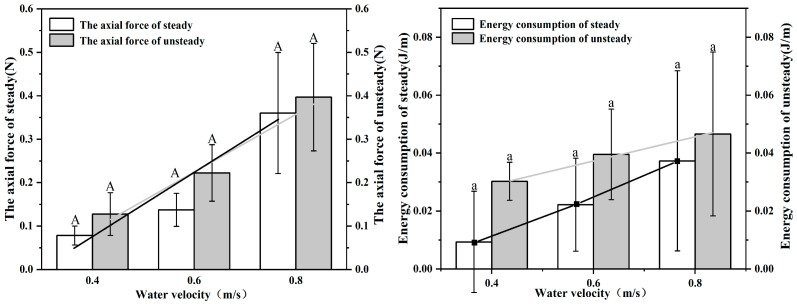
Relationship between energy consumption and water flow velocity in steady/unsteady mode of *S. malacanthus.* Different upper cases indicate the significant the difference of water velocity in the axial force of steady/unsteady treatments; Different lower cases indicate the significant the difference of water velocity in energy consumption of steady/unsteady treatments.

**Figure 7 animals-12-02844-f007:**
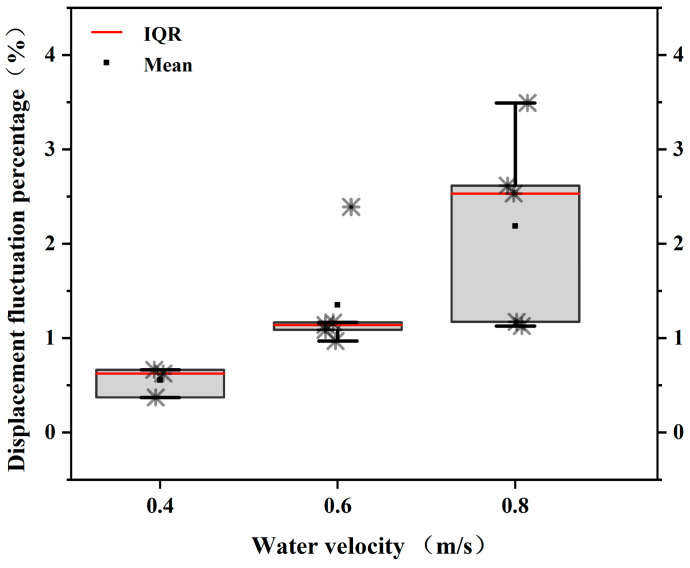
Relationship between thrust and water flow velocity in steady/unsteady mode of *Schizopygopsis malacanthus* swimming. Note: IQR: Interquartile range.

**Table 1 animals-12-02844-t001:** Flow velocities for the stepped velocity test.

Test Fish	Flow Velocity (m/s)			
1	0.4	0.6	0.8	1.0
2	0.4	0.6	0.8	1.0
3	0.4	0.6	0.8	-
4	0.4	0.6	0.8	-
5	0.4	0.6	0.8	1.0

**Table 2 animals-12-02844-t002:** List of symbols used in the kinematic analyses of *S. malacanthus* swimming.

Physical Indicator	Definition	Unit
$\bar{S}$	Average instantaneous displacements	cm
σ	Standard deviation of instantaneous displacements	cm
ρ	Water density	kg/m^3^
$A_{s}$	Wetted surface area of fish body	m^2^
$U_{i}$	Absolute swimming speed of fish	m/s
U	Mean swimming speed	m/s
ϑ	Kinematic viscosity of water	m^2^/s
*L*	Total length of fish	cm
T	Temperature of flume	°C
$C_{dc}$	Forward drag coefficient	-
$C_{ds}$	Backward drag coefficient	-
$R_{ec}$	Forward Reynolds number	-
$R_{es}$	Backward Reynolds number	-
$v_{w}$ $v_{c}$	Water velocity Fish forward velocity	m/s m/s
$v_{s}$	Fish backward velocity	m/s
$C_{A}$	Added mass coefficient	-
$L_{f}$	Fish body length	cm
m	Fish mass	g
$a_{c}$ $a_{s}$	Forward acceleration	m/s^2^
	Backward acceleration	m/s^2^
$C_{f}$	Friction coefficient	-
$C_{p}$	Pressure coefficient	-
$t_{f}$	Unsteady mode time	s
$t_{w}$	Steady mode time	s
$t_{i}$	Unsteady or steady swimming time	s
$U_{max}$	Maximum swimming speed	m/s
$a_{max}$	Maximum acceleration	m/s^2^
f	Force to overcome drag	N
F	Force required for acceleration	N
$F_{total}$	Total force	N
$E_{total}$	Total energy consumption	J/m

Note: “-” denotes a dimensionless quantity.

**Table 3 animals-12-02844-t003:** Body Length and $U_{crit}$ for each test fish.

Test Fish	Body Length (cm)	$U_{crit}\;\;(m/s)$
1	18.3	1.01
2	14.3	0.97
3	18.2	0.80
4	15.0	0.69
5	16.3	0.81
Mean ± SE (*n* = 5)	16.4 ± 0.8	0.86 ± 0.06

**Table 4 animals-12-02844-t004:** Physical variables used to analyze *S. malacanthus* swimming.

Mode	Physical Indicator	0.4	0.6	0.8
-	$\bar{S}$ (cm)	0.011 ± 0.013	0.005 ± 0.007	0.031 ± 0.031
-	σ (cm)	0.341 ± 0.129	0.465 ± 0.067	0.788 ± 0.159
Steady	$t_{w}$ (%) $t_{s}$ (%)	85.07 ± 0.114	77.28 ± 0.105	70.04 ± 0.034
Unsteady		14.93 ± 0.114	22.72 ± 0.105	29.96 ± 0.034
Steady	$v_{c}$ (m/s^2^)	0.022 ± 0.005	0.059 ± 0.011	0.084 ± 0.022
Unsteady		0.078 ± 0.012	0.159 ± 0.026	0.194 ± 0.051
Steady Unsteady	$v_{s}$ (m/s^2^)	−0.021 ± 0.003	−0.056 ± 0.008	−0.083 ± 0.021
		−0.105 ± 0.016	−0.146 ± 0.031	−0.186 ± 0.049
Steady Unsteady	$a_{c}$ (m/s^2^)	0.9 ± 0.096	2.381 ± 0.420	3.012 ± 0.684
		2.012 ± 0.28	3.580 ± 0.755	5.079 ± 1.634
Steady Unsteady	$a_{s}$ (m/s^2^)	−0.797 ± 0.089	−2.365 ± 0.479	−3.203 ± 0.734
		−2.258 ± 0.368	−3.628 ± 0.651	−4.920 ± 1.590
Steady Unsteady	$F_{total}$(N)	0.078 ± 0.022	0.137 ± 0.038	0.340 ± 0.139
		0.128 ± 0.049	0.222 ± 0.065	0.397 ± 0.124
Steady Unsteady	$E_{total}$ (J/m)	0.009 ± 0.007	0.022 ± 0.016	0.037 ± 0.028
		0.03 ± 0.018	0.04 ± 0.016	0.047 ± 0.031
Unsteady	$U_{max}$ (m/s)	0.613 ± 0.025	0.973 ± 0.078	1.418 ± 0.164
	$a_{max}$ (m/s^2^)	11.216 ± 5.758	11.736 ± 3.148	16.540 ± 5.671

Note: Please see Table 2 for the definition of symbols in this table.

## Data Availability

The data presented in this study are available on request from the corresponding author.

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
