# Peer review of "The Kinematics and Dynamics of Schizopygopsis malacanthus Swimming during Ucrit Testing"

_animals, 2022, doi:10.3390/ani12202844_

Round 1

Reviewer 1 Report

This manuscript is a fascinating, well-written article, and I think it can be accepted with minor revisions. In this paper, the motion of Schizopygopsis malacanthus during the Ucrit test is analyzed frame by frame through the video analysis and tracking software Ethovision XT, and the kinematics and dynamics indexes under each flow velocity gradient when Ucrit is reached are obtained. The research content of the article is rich, novel, clear thinking, standard writing, in line with the publication requirements of this journal, but there are still some small problems.

(1)   In Section 2.4, how to determine the specific location of the centroid? Is it the middle of the head or the middle of the body? 

(2)   In Section 2.6, is the axial force transverse or longitudinal?

(3)   In Discussion 4.1, is Wise defined like this? I feel that the original text is unlikely to be defined this way. The fluctuation is that the tail swing exceeds 2%, which means unstable swimming. Or is the difference between 2 tail swings greater than 2% and one is an unstable swim?

(4)   Please add more details about steady time and unsteady time. How are they defined and calculated?

Author Response

Dear editors and experts

     Thank you for your review of this paper. Each of your comments is very pertinent and will be of great help to me and my co-authors both in this paper and in future scientific research. After our unremitting efforts, the revisions are marked in red font in the article. We also sincerely hope that the revised paper can meet your requirements.

All the best!

The specific reply to the comments of the second reviewer is as follows.

  1. In Section 2.4, how to determine the specific location of the centroid? Is it the middle of the head or the middle of the body?

A:Thank reviewers for their comments. We use trajectory analysis software (Etho Vision XT 9.0) to analyze fish, and the software determines the middle of the fish body, which is also the definition of the general middle of mass.

  1. In Section 2.6, is the axial force transverse or longitudinal?

A: Thank reviewers for their comments. The axial force is in the direction of the fish body, that is, longitudinal.

  1. In Discussion 4.1, is Wise defined like this? I feel that the original text is unlikely to be defined in this way. The fluctuation is that the tail swing exceeds 2%, which means unstable swimming? Or is the difference between 2 tail swings greater than 2% and one is an unstable swim?

A: Thank reviewers for their comments. The fluctuation defined by Wise is that the displacement of fish moving forward does not exceed 2% of its body length, but the difference between the two tail swings exceeds 2%.

  1. Pleaseaddmore details about steady time and unsteady time. How are they defined and calculated?

A: Thank reviewers for their comments. We introduce the definitions of steady tw and unsteady time tf in 2.4. The steady time is defined as the time period when the displacement is within the standard deviation as the steady time, and the time period when the displacement is outside the standard deviation as the unsteady time; The steady and unsteady time can be calculated by directly accumulating their time periods.

Reviewer 2 Report

The manuscript “The Kinematics and Dynamics of Schizopygopsis malacanthus Swimming During Ucrit Testing” describes the kinematics of steady and unsteady swimming of a riverine fish species in a flow tank at incremental speeds. The fish exhibited both steady and unsteady swimming in constant flow conditions, body length was only significant for speed and acceleration in steady swimming, and burst-coast behaviors were found to increase energy consumption instead of decrease it. These results were then used to suggest vertical slit fishways for future flume experiments. Overall, the paper has merit, but there are a few areas of improvement to address, especially some areas that are unclear or need to be expanded. Therefore, I recommend minor edits before potential publication as an Article in Animals. More detailed comments are listed below:

1.     In the abstract, lines 24-26, “defining steady and unsteady swimming statistically, in terms of the standard deviation of instantaneous displacements, is an improvement,” is a confusing statement. I think this is clarified later in the discussion, line 305. However, I suggest emphasizing this definition in the discussion and clarifying this in the abstract.

2.     Ucrit is defined well in the second paragraph but is introduced in the first paragraph before this actual definition. I suggest restructuring this so that the definition comes first, or at least doesn’t come that much later than the word is first used.

3.     Similarly, the overall introduction is quite short. Can you expand upon the previous findings and explain more about the choice for this fish species? Information about the fish, including an image, is helpful to the reader.

4.     In line 54, “most sophisticated style of swimming” is inaccurate. Instead, change to “complicated,” “advanced,” or something that does not have the same connotations as “sophisticated.”

5.     Does the range of temperatures affect swimming performance? Plot the results by temperature and include in a supplemental file.

6.     The use of energy consumption is confusing because there is also metabolic energy that could be directly measured (e.g., using an oxygen probe or other instruments). There should be at least a few sentences clarifying this in the methods and discussion to address the definition of energy consumption and its components, and how/why metabolism is not explicitly used in the equation.

7.     In Table 1, why is the test fish 1 and first two velocities bolded and underlined? It seems like an error, or if this is supposed to be highlighted then include more information in the caption.

8.     In Figure 1, include dimensions and scale bars in the image. There is also no information in the caption, beyond that this is a swim tunnel. Label the components.

9.     In Table 2, kinematic viscosity should not unitless. Double check the term and units you used.

10.  In Figure 6, define IQR in the caption.

11.  In lines 330-333, what is the source for this statement on fish body shape and acceleration/agility?

12.  There are numbers throughout the text, presumably referring to the numbered sources. Please fix the formatting.

Author Response

Dear editors and experts

    Thank you for your review of this paper. Each of your comments is very pertinent and will be of great help to me and my co-authors both in this paper and in future scientific research. After our unremitting efforts, the revisions are marked in red font in the article. We also sincerely hope that the revised paper can meet your requirements.

All the best!

The specific reply to the comments of the first reviewer is as follows.

  1. In the abstract, lines 24-26, "defining steady and unsteady swimming statistically, in terms of the standard deviation of instantaneous displacements, is an improvement," is a confusing statement. I think this is clarified later in the discussion, line 305. However, I suggest emphasizing this definition in the discussion and clarifying this in the abstract.

A:Thank reviewers for their comments. We have simply modified the abstract.

  1. Ucrit is defined well in the second paragraph but is introduced in the first paragraph before this actual definition. I suggest restructuring this so that the definition comes first, or at least doesn't come that much later than the word is first used.

A:Thank reviewers for their comments. As for the description of Ucrit, in fact, the first paragraph describes the shortcomings of Ucrit research—kinematics and dynamics are rarely studied. In the second paragraph, what Ucrit generally refers to in kinematics is introduced, not the definition of Ucrit. Therefore, after discussion, the authors chose not to change it.

  1. Similarly, the overall introduction is quite short. Can you expand upon the previous findings and explain more about the choice for this fish species? Information about the fish, including an image, is helpful to the reader.

A:Thank reviewers for their comments. The reason for choosing the experimental fish has been explained in the introduction of the article, and the photos of the experimental fish have been added to the article.

  1. In line 54. "most sophisticated style of swimming" is inaccurate. Instead, change to"complicated." "advanced," or something that does not have the same connotations as sophisticated.

A:Thank reviewers for their comments. We have changed "sophisticated" to "replicated".

  1. Does the range of temperatures affect swimming performance? Plot the results by temperature and include in a supplemental file.

A: General temperature will affect the swimming ability of fish [Jain KE, Farrell AP. Influence of seasonal temperature on the repeat swimming performance of rainbow trout Oncorhynchus mykiss[J]. Journal of Experimental], but in this paper, the water temperature in the experimental tank is consistent with the temperature of fish in the natural environment.

  1. The use of energy consumption is confusing because there is also metabolic energy that could be directly measured (e.g., using an oxygen probe or other instruments). There should be at least a few sentences clarifying this in the methods and discussion to address the definition of energy consumption and its components, and how/why metabolism is not explicitly used in the equation.

A:Thank reviewers for their comments. We have explained and explained in Section 2.6.

  1. In Table 1, why is the test fish 1 and first two velocities bolded and underlined? It seems like an error, or if this is supposed to be highlighted then include more information in the caption.

A: Thank reviewers for their comments. This is because of our mistakes, and we have changed it.

  1. In Figure 1, include dimensions and scale bars in the image. There is also no information in the caption, beyond that this is a swim tunnel. Label the components.

A:Thank reviewers for their comments. The dimensions in the figure have been explained in the text, and this is produced by Loligo Company, which is internationally recognized.

  1. In Table 2, kinematic viscosity should not unitless. Double check the term and units you used.

A:Thank reviewers for their comments. We added the unit of viscosity coefficient "m2/s" in Table 2 and checked the unit of the full text.

  1. In Figure 6, define IQR in the caption.

A:Thank reviewers for their comments. IQR: Interquartile Range has been defined in the text.

  1. 1 In lines 330-333, what is the source for this statement on fish body shape and acceleration/agility?

A:Thank reviewers for their comments. We express that in addition to the impact of water flow speed on the acceleration of fish, fish species and body length may also have an impact. The following statements are examples of cited literature.

  1. There are numbers throughout the text, presumably referring to the numbered sources. Please fix the formatting.

A:Thank reviewers for their comments. Numbers refer to references.
